# Epidemiology and Outcome of Pediatric Out-of-Hospital Cardiac Arrest after Traffic Collision in Japan: A Population-Based Study

**DOI:** 10.3390/jcm11030831

**Published:** 2022-02-04

**Authors:** Sanae Hosomi, Tetsuhisa Kitamura, Tomotaka Sobue, Ling Zha, Kosuke Kiyohara, Jun Oda

**Affiliations:** 1Department of Traumatology and Acute Critical Medicine, Graduate School of Medicine, Osaka University, 2-15, Yamada-oka, Suita 565-0871, Japan; s-hosomi@hp-emerg.med.osaka-u.ac.jp (S.H.); odajun@gmail.com (J.O.); 2Division of Environmental Medicine and Population Sciences, Department of Social Medicine, Graduate School of Medicine, Osaka University, 2-2 Yamadaoka, Suita 565-0871, Japan; tsobue@envi.med.osaka-u.ac.jp (T.S.); ivy_mist@outlook.com (L.Z.); 3Department of Food Science, Faculty of Home Economics, Otsuma Women’s University, 12 Sanban-cho, Chiyoda-ku, Tokyo 102-8357, Japan; kiyosuke0817@hotmail.com

**Keywords:** traffic collision, mortality, out-of-hospital cardiac arrest, trauma, Japan

## Abstract

The epidemiological and clinical characteristics, treatments, and outcomes of patients with traumatic out-of-hospital cardiac arrests (OHCAs) following traffic collisions have not been adequately investigated in Japan. We analyzed the All-Japan Utstein Registry data of 918 pediatric patients aged <20 years with OHCAs following traffic collisions who were resuscitated by bystanders or emergency medical service personnel and were subsequently transported to hospitals between 2013 and 2019. Multiple logistic regression analysis was used to assess factors potentially associated with 1-month survival after OHCA. The 1-month survival rate was 3.3% (30/918), and the rate of neurologically favorable outcomes was 0.7% (60/918). The proportion of 1-month survival of all OHCAs after traffic collision origin did not significantly increase (from 1.9% (3/162) in 2013 to 4.5% (5/111) in 2019), and the adjusted odds ratio (OR) for a 1-year increment was 1.13 (95% confidence interval (CI) 0.93 to 1.37). In a multivariate analysis, ventricular fibrillation arrests and pulseless electrical activity (PEA) were significant predictors of 1-month outcome after OHCAs due to traffic collision. From a large OHCA registry in Japan, we demonstrated that 1-month survival after OHCAs due to traffic collision origin was approximately 3%, and some children even gained full recovery of neurological function.

## 1. Introduction

Data from the field of pediatrics show that trauma is the leading cause of death in industrialized countries, with traffic collisions being the main cause of traumatic death worldwide [1,2,3,4]. A large group of patients do not survive traumatic out-of-hospital cardiac arrest (OHCA) despite appropriate, immediate, and extensive efforts by bystanders and emergency medical service (EMS) personnel. These patients present challenges to the healthcare system because prehospital resuscitative efforts, as well as rescue attempts in hospital settings, are resource and labor intensive.

The 2010 Consensus on Science with Treatment Recommendations (CoSTR) recommended the use of validated termination of resuscitation rules in adults for all OHCAs, including non-traumatic OHCA cases [5]. In 2020, the Education, Implementation, and Teams (EIT) Task Force softened this to a conditional recommendation, taking into consideration the social unacceptability of excluding potential survivors from in-hospital treatment and due to a lack of supporting evidence.

The etiology of cardiac arrest following a traffic collision is mostly assumed to be blunt trauma. According to the joint position statement of the National Association of EMS Physicians and the American College of Surgeons Committee on Trauma (ACS-COT) [1], the resuscitation of patients with either blunt or penetrating trauma, who have no detectable pulse, are apneic, and do not show signs of life is not recommended, and they are to be pronounced dead at the scene of injury. However, a review reported that survival after blunt trauma was significantly higher in the pediatric group than in the adult group [4]. There are no specific guidelines regarding the withdrawal of treatment for children or adolescents with cardiac arrest secondary to trauma. Thus, the management of traumatic OHCA in pediatric patients should be handled carefully, in tandem with the continuous monitoring of mortality prediction values.

The All-Japan Utstein Registry is a large prospective population-based cohort study of OHCAs in Japan. It was launched in 2005 and involves approximately 127 million residents [6,7,8]. In 2013, we began the evaluation of detailed causes of non-cardiac OHCA. Using this nationwide database, we evaluated the epidemiology and outcomes of pediatric traumatic OHCA following traffic collisions in Japan. The government made some investments in road traffic safety through the Vision Zero policy [9]. As a result, many automakers are working hard to develop safer motor vehicles and traffic systems. Despite these efforts, accidents still occur. Pediatric traumatic OHCA following traffic collision has not been fully investigated at the national level, and it is important to understand the actual trend.

## 2. Materials and Methods

### 2.1. Study Design and Setting

The All-Japan Utstein Registry is a prospective, population-based registry of OHCAs, and it is based on the standardized Utstein style [10,11]. This study enrolled pediatric patients aged <20 years (the age of adulthood in Japan) who experienced OHCAs after traffic collision and before EMS arrival, who were resuscitated by bystanders or EMS personnel, and who were transported to hospitals in Japan between 1 January 2013 and 31 December 2019. In this study, we excluded adult patients with OHCA because the characteristics and outcomes of OHCAs differ between pediatrics and adults [12,13].

Cardiac arrest was defined as the cessation of cardiac mechanical activity, as confirmed by the absence of signs of circulation [6,7,8]. In this registry, the arrests are classified into cases of presumed cardiac origin and those of non-cardiac origin, the latter resulting from cerebrovascular disease, asphyxia, malignant tumors, external causes, drawing, drug overuse, accidental hypothermia, traffic collision, and other causes. These diagnoses were made clinically by the physician responsible for treatment, working in cooperation with EMS personnel. Patients with traumatic OHCAs following a traffic collision are included.

### 2.2. EMS Organization in Japan

Japan has an area of approximately 378,000 km^2^, including both urban and rural communities, with a population of 126 million inhabitants. Twenty million, seven hundred thousand of these inhabitants were younger than 20 years of age according to the 2020 Census [14,15]. There were 726 fire stations with dispatch centers in 2021, and their EMS systems were almost uniform [15]. Details of the EMS system in Japan have been described previously [4]. The EMS system is operated by local fire stations. When contacted, an ambulance is dispatched from the nearest fire station. Emergency services are provided 24 h a day, every day. The most highly trained prehospital emergency care providers are called emergency life-saving technicians (ELSTs). Usually, each ambulance has a crew of three emergency providers, including at least one ELST. They are allowed to insert an intravenous line and an adjunct airway and to use a semiautomated external defibrillator for patients with OHCA. Since July 2004, specially trained ELSTs have been permitted to perform tracheal intubation, and since April 2006, they have been permitted to administer intravenous epinephrine. Do-not-resuscitate orders or living wills are generally not accepted in Japan. EMS providers are not permitted to terminate resuscitation at the scene. Therefore, almost all patients with OHCA who are treated by EMS personnel are transported to a hospital and enrolled in the All-Japan Utstein Project, excluding those with decapitation, incineration, decomposition, rigor mortis, or livor mortis.

The use of an automated external defibrillator (AED) by citizens has been legally permitted since July 2004. In Japan, approximately 2 million citizens participate in community cardiopulmonary resuscitation (CPR) programs, which include training in chest compression, mouth-to-mouth ventilation, and the use of AED [6,7,8]. All EMS providers perform CPR according to the Japanese CPR guidelines [16].

### 2.3. Data Collection and Quality Control

Data were prospectively collected using a form that included data recommended in the Utstein-style reporting guidelines for cardiac arrests [10,11]. Data regarding patient age, sex, type of bystander witness status, first recorded cardiac rhythm, life support by EMS personnel (i.e., use of advanced life support (ALS) devices or the insertion of an intravenous line), time course of resuscitation and epinephrine administration, prehospital return of spontaneous circulation (ROSC), and 1-month survival were obtained. Data regarding time of call to EMS, arrival of the ambulance at the scene of the accident, contact with patients, initiation of CPR, defibrillation performed by EMS personnel, and arrival at the hospital were recorded using the clock of each EMS system. In cases of shock delivery by bystanders using a public-access AED, the patient’s first recorded rhythm was regarded as ventricular fibrillation or pulseless ventricular tachycardia. Information about the type of CPR performed by a bystander was obtained through observation and interviews carried out with the bystander by EMS personnel before leaving the scene of the accident. The data forms were completed by EMS personnel in collaboration with the physicians in charge. The data were integrated into the All-Japan Utstein Registry database server and logically checked by the computer system. In cases of incomplete data forms, the Fire and Disaster Management Agency requested the provision of missing data from the respective fire stations.

All survivors who had suffered an OHCA were followed up for up to 1 month after the event by the EMS personnel in charge. One-month neurological outcomes were determined by the physician in charge of treating the patient using the cerebral performance category (CPC) scale: category 1, good cerebral performance; category 2, moderate cerebral disability; category 3, severe cerebral disability; category 4, coma or vegetative state; and category 5, death [10,11].

### 2.4. Outcome Measures

The main outcome measure was 1-month survival. Secondary outcome measures included prehospital ROSC and 1-month survival with neurologically favorable outcomes. Neurologically favorable outcomes were defined as CPC category 1 or 2 [10,11].

### 2.5. Statistical Analysis

Categorical variables were presented as counts with proportions, and the χ2 test was used to evaluate differences between the two groups. Continuous variables were presented as medians with interquartile ranges (IQRs), and the Wilcoxon Mann–Whitney test was used to evaluate differences between the two groups.

The age-standardized annual incidence of OHCAs after traffic collision was calculated using the direct method, using 2013–2019 population data from the Statistics Bureau of Japan and the 1985 Japanese model population [17,18]. The annual trends were assessed using linear trend tests. Multiple logistic regression analysis was used to assess factors associated with 1-month survival, prehospital ROSC, and neurologically favorable outcomes, and adjusted odds ratios (ORs) and their 95% confidence intervals (CIs) were calculated. As potential confounders, factors that were biologically essential and considered to be associated with clinical outcomes were included in the multivariable analyses [8,13,14]. These variables included age (<1, 2–5, 6–11, ≥12 years old), sex (male, female), witness status (none or witnessed by bystanders), first documented rhythm (ventricular fibrillation (VF)/pulseless ventricular tachycardia (VT), pulseless electrical activity (PEA), asystole), bystander CPR status (any CPR given or no CPR given), advanced airway management (AAM) (endotracheal intubation (ETI), supraglottic airway (SGA), or no AAM), intravascular fluid (yes or no), epinephrine (yes or no), EMS response time (from the call to contact with patients), time taken between contact with patients and arrival to hospital, daytime (AM 9:00–PM 4:59) (yes or no), weekend/holiday (yes or no), and year of arrest. For the multivariable analyses of prehospital ROSC, all variables except for contact with patients to hospital arrival were used because they did not affect prehospital ROSC.

All statistical analyses were conducted using STATA (v16; StataCorp LP, College Station, TX, USA). All tests were two-tailed, and *p* values < 0.05 were considered statistically significant.

### 2.6. Ethics Approval

This manuscript was written based on the STROBE statement for the reporting of cohort and cross-sectional studies [19]. The study design was approved by the Ethics Committee of the Osaka University Graduate School of Medicine (approval number: 14147). The requirement for written informed consent was waived owing to the retrospective nature of the study. Personal identifiers are not included in the Utstein records.

## 3. Results

### 3.1. Eligible Patients

Figure 1 shows an overview of the study patients based on the Utstein template. A total of 12,843 pediatric cardiac arrests were documented during the 7 years. Resuscitation was attempted in 11,921 cases. Excluding 698 patients whose arrests were witnessed by EMS (arrests after EMS arrival) and 79 unknown, 11,144 (3389 bystander-witnessed cases and 7755 non-witnessed cases) remained. Among these arrests, 944 were due to traffic collisions. We could not obtain information on the first cardiac rhythm in 26 (2.75%) patients. The remaining 918 patients were eligible for the study.

### 3.2. Annual Incidence of Traumatic OHCA Following Traffic Collision

The age-standardized annual incidence rates per 100,000 persons of traumatic OHCA following traffic collision were calculated over time (Figure 2). The incidence rate of traumatic OHCA following traffic collision remained largely unchanged from 2013 (0.198) to 2019 (0.143; *p* for trend = 0.071).

### 3.3. Description of Baseline Features

Patient and EMS characteristics of traumatic OHCA following traffic collision and their outcomes are shown in Table 1. Of the 918 patients with traumatic OHCA following traffic collision, 7 (0.76%) patients were under 1 years old, 129 (14.05%) were 1–5, 126 (13.73%) were 6–11, and 656 (71.46%) were 12–19. Although the proportion varied among the four groups, most of the patients were boys (683/918, 74.4 %) and had witnesses (670/918, 73.0%). The first rhythm was commonly the nonshockable rhythm (pulseless electrical activity and asystole), and the VF rhythm was only 1.9% (17/918).

As age increased, the frequency of bystander CPR decreased (4/7 (57.1%) for the <1 age group, 56/129 (43.4%) for the 1–5 age group, 52/126 (41.3%) for the 6–11 age group, and 159/656 (24.2%) for the 12–19 age group; *p* < 0.001), and the rate of intravascular fluid and epinephrine increased (intravascular fluid: 0/7 (0%) in the <1 age group, 2/129 (1.6%) in the 1–5 age group, 13/126 (10.3%) in the 6–11 age group, and 147/656 (22.4%) in the 12–19 age group: epinephrine; 0/7 (0%) in the <1 age group, 2/129 (1.6%) in the 1–5 age group, 8/126 (6.3%) in the 6–11 age group, and 102/656 (15.5%) in the 12–19 age group, *p* < 0.001). The proportion of patients with advanced airway management was 0% for both ETI and SGA in the <1 group, 2.7% in the ETA group, and 23.0% in the SGA of the 12–19 group. Although most cases of traumatic OHCA following traffic collision were at nighttime and weekends/holidays, the detailed rates varied between younger and older children (daytime: 4/7 (57.1%) in the <1 age group, 78/129 (60.5%) in the 1–5 age group, 68/126 (54.0%) in the 6–11 age group, and 128/656 (21.0%) in the 12–19 age group; weekend/holiday: 3/7 (42.9%) in the <1 age group, 74/129 (57.4%) in the 1–5 age group, 96/126 (76.2%) in the 6–11 age group, and 450/656 (68.6%) in the 12–19 age group).

### 3.4. Description of Outcomes

In all patients, the rates of 1-month survival, prehospital ROSC, and neurologically favorable outcomes were 3.3% (30/918), 6.5% (60/918), and 0.7% (60/918), respectively. The proportion of 1-month survival of traumatic OHCA following traffic collision did not significantly vary among age groups.

The trends for outcomes by year are shown in Table 2. The adjusted ORs for a 1-year increment in 1-month survival, prehospital ROSC, and neurologically favorable outcomes after traumatic OHCA following traffic collision were almost similar (adjusted OR 1.13, 95% CI 0.93 to 1.37, adjusted OR 1.03, 95% CI 0.90 to 1.17, and adjusted OR 1.28, 95% CI 0.83 to 1.97, respectively).

### 3.5. Factors Related to Mortality

Table 3 shows the factors that contributed to 1-month survival after traumatic OHCA following traffic collision. In 1-month survival, VF as the first documented rhythm (adjusted OR 12.15, 95% CI 2.04 to 72.36) and PEA as the first documented rhythm (adjusted OR 4.46, 95% CI 1.82 to 10.93) were associated with improved outcome. However, bystander CPR (adjusted OR 0.84, 95% CI 0.36 to 1.98), intravenous fluid levels (adjusted OR 0.51, 95% CI 0.08 to 3.28), and AAM by SGA (adjusted OR 1.32, 95% CI 0.46 to 3.80) were not associated with outcome.

Table 4 shows the factors that contributed to prehospital OHCA after traumatic OHCA following traffic collision. The results are similar to those presented in Table 3.

## 4. Discussion

The extensive OHCA registry in Japan showed that 1-month survival after traumatic OHCA following traffic collision was approximately 3%, and the survival trends did not improve as the years went by. However, some children gained full recovery of their neurological function. In order to improve survival after OHCAs, additional attention should be paid to the epidemiological characteristics of traumatic OHCA following traffic collision as is paid to OHCAs of cardiac origin. This study describes the actual situation regarding the incidence and outcome of pediatric traumatic OHCAs following traffic collision, providing valuable information to improve survival.

The 2016 rate of death from traffic collision among children and adolescents from the U.S. was 5.21 per 100,000 (95% CI, 5.06 to 5.38) [20]. This rate was 2.94 per 100,000 (95% CI, 2.52 to 3.43) in Australia, 1.04 per 100,000 (95% CI, 0.87 to 1.23) in England and Wales, and 0.91 per 100,00) (95% CI, 0.56 to 1.45) in Sweden. Traffic collision is still the leading cause of death in children and adolescents, even in high-income countries [20]. Among all traffic collision-related deaths, the most serious cases involve traumatic OHCA. Our study shows that the rate of death from traumatic OHCA following traffic collision in Japan is less than 0.20 per 100,000.

Although trauma-related deaths have traditionally been viewed as “accidents”, injury prevention science, which was developed in the latter part of the 20th century, increasingly shows that deaths due to trauma are preventable with empirical evidence. To reduce traumatic deaths caused by motor vehicle collisions, the Japanese Road Traffic Act was revised in June 2002 and imposed severe fines for traffic offenses. As a result, fatal collisions have decreased since then. Additionally, the law mandates that children aged <6 years need to use child seats. In fact, deaths due to traffic collisions have dramatically decreased in Japan [21]. However, our study shows that the outcomes of pediatric traumatic OHCA following traffic collision have not improved in recent years. Thus, further efforts to reduce pediatric mortality are needed. It is not possible to obtain a driver’s license until one is 16 years old or older under Japanese law; therefore, these victims under 15 years old cannot be “driving themselves”, meaning that parents or caregivers are mainly responsible for protecting infants and children. The improvement in Japan might be attributed to the widespread adoption of seat belts and appropriate child safety seats, the production of cars with improved safety standards, better constructed roads, and graduated driver licensing programs as reported in the U.S. [22,23].

Considering our results on patient background, the proportion of bystander CPR for traumatic OHCAs following traffic collision in Japan might be lower than that of OHCAs of non-cardiac origin [8,13]. There might be differences in the proficiency of bystander basic life support (BLS) procedures between cardiac and traumatic OHCA. In such serious trauma cases as cardiac arrest at the scene, the effectiveness of BLS might be limited clinically. As recommended in the CPR guidelines [16], the activation of EMS plays a key role in the “chain of survival”.

Injured OHCA pediatric patients in prehospital settings may be a different population to adult patients in similar circumstances and justify a more aggressive initial resuscitative effort [14]. It was reported that cardiopulmonary arrest (CPA) in pediatric patients may not be a result of hemorrhage shock but of a primary respiratory cause. Children are especially prone to CPA due to respiratory failure [23]. Simple airway maneuvers and intubations may prevent the development of hypoxic cardiac arrest. Other deaths following trauma in the acute phase might be due to hemorrhagic shock, and uncontrolled bleeding is the primary cause of death in up to one-third of patients who suffer trauma-related death [20]. However, patients experiencing bleeding associated with hypovolemic arrest at the scene of injury are unlikely to respond to conventional CPR. The 2020 CoSTR for PLS referred to the importance of volume resuscitation for pediatric victims with traumatic hemorrhagic shock, as well as the management of children with cardiac arrest following trauma [20]. Considering the low proportion of prehospital ALS procedures in our study, more aggressive ALS might improve the mortality of pediatric traumatic OHCA following traffic collision. Further studies to improve mortality from pediatric traumatic OHCA are needed.

In our study, PEA and VF/pVT were independent predictors of favorable outcomes after traumatic OHCA following traffic collision in the multivariate analysis. These results are contrary to the report that the initial rhythm in the case of trauma is probably not a potent predictor of patient outcome [21], or that no patient with VF/tachycardia survived [24], which might be due to the difficulty in accurately analyzing pediatric pulses in prehospital settings without AED auto-analysis, leading to some bias.

It is usually assumed that traumatic OHCAs in the prehospital setting result in undesirable outcomes, leading physicians to view cardiopulmonary resuscitation in such cardiac arrests as ineffective [25]. It is debated whether the number of survivors is too limited, considering the risks of resuscitation efforts and inherent costs. Simultaneously, some studies showed survivors being resuscitated despite meeting the criteria for cessation or the withholding of treatment according to the guidelines [26]. It was concluded that assessments in the prehospital setting may not be effective in determining the need for triage of trauma patients with OHCA and that they should be transported to a medical institution for further evaluation and care [26]. Furthermore, the withdrawal of CPR in pediatric cases is a difficult and emotionally charged issue [24]. Medical personnel caring for pediatric patients with a traumatic OHCA are faced with enormous conflict, and they must make a decision within only a few minutes about when and if to stop resuscitation efforts while also considering the high percentage of mortality [27] and probable poor neurological outcomes in survivors [28]. However, it was reported that pediatric patients have a higher possibility of survival after resuscitation of traumatic OHCA than adults [4]. In fact, in our study, some patients in the older age group even gained full recovery of neurological function. The mentioned guidelines should not be used for pediatric patients, since the literature suggests improved survival for children; moreover, there is very limited clinical validation [29]. Our results could help in the decision making of medical teams who work not only in prehospital settings but also in hospitals when traumatic OHCA following traffic collision occurs. Therefore, this study provides recommendations for future guidelines.

This study has some inherent limitations. The Utstein-based data utilized in this study did not obtain the quality of out-of-hospital CPR; specific clinical information about the injuries; or in-hospital data, such as post-arrest care [30]. We do not know whether the victims were pedestrians, cyclists, motorcyclists, or drivers/passengers. Furthermore, the autopsy data were not obtainable because our data were collected in accordance with the Utstein-style guidelines by EMS personnel. Therefore, our study could not identify the detailed origin of the presumed OHCAs nor estimate OHCAs due to a medical origin in those caused by traffic injuries. Second, our results might not be fully capable of being applied to other parts of the world, including the U.S. and Europe, which have different EMS systems. Therefore, further investigations of other cohorts are warranted to confirm these associations and to address generalizability. Third, there might be unmeasured confounding factors that might have influenced the association between traumatic OHCA following traffic collision and outcomes. This study does not include data during the COVID-19 pandemic period, but this may change during a longer observation period [27]. Finally, as with all epidemiological studies, data integrity, ascertainment, and validity bias are potential limitations. The use of uniform data collection based on Utstein-style guidelines for reporting cardiac arrest, a large sample size, and a population-based design to involve all known OHCAs in Japan was intended to minimize these potential sources of bias.

## 5. Conclusions

The large OHCA registry in Japan demonstrates that 1-month survival after traumatic OHCAs following traffic collision is poor, and the survival trends have not improved year by year. However, some children gain full recovery of neurological function. Research related to better safety equipment for children and enhanced training for pediatric-related trauma are warranted to improve survival outcomes after OHCA in this group.

## Figures and Tables

**Figure 1 jcm-11-00831-f001:**
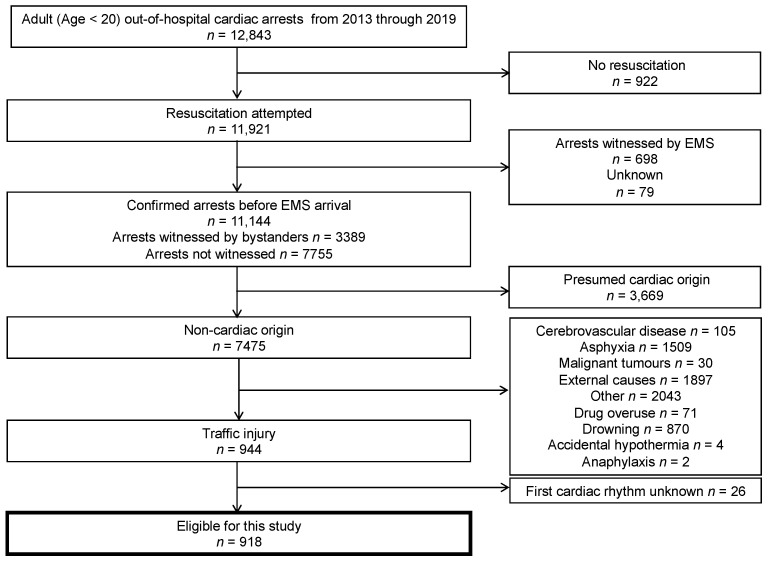
Flowchart depicting the selection of the study population. EMS = emergency medical service.

**Figure 2 jcm-11-00831-f002:**
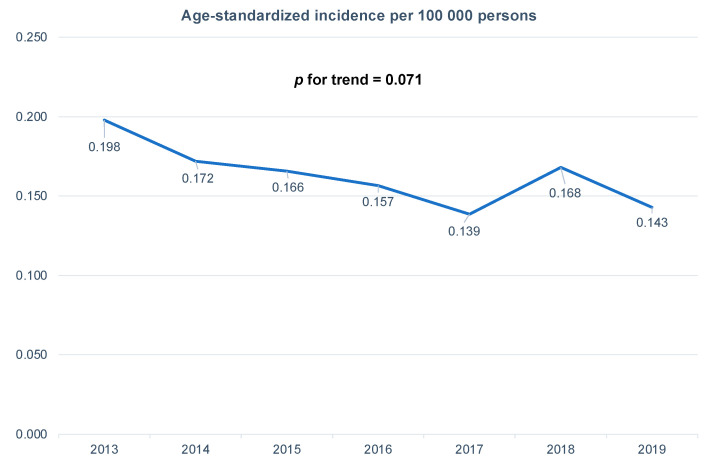
Age-standardized incidences of traumatic out-of-hospital cardiac arrests following traffic collisions from 2013–2019.

**Table 1 jcm-11-00831-t001:** Patient characteristics and outcomes of traumatic out-of-hospital cardiac arrests after traffic accidents by age group.

		Total	<1	1–5	6–11	12–19	
		*n* = 918	*n* = 7	*n* = 129	*n* = 126	*n* = 656	*p*-Value
Year of onset	2013	162 (17.6%)	0 (0.0%)	25 (19.4%)	21 (16.7%)	116 (17.7%)	0.810
	2014	140 (15.3%)	0 (0.0%)	15 (11.6%)	24 (19.0%)	101 (15.4%)	
	2015	134 (14.6%)	1 (14.3%)	21 (16.3%)	19 (15.1%)	93 (14.2%)	
	2016	127 (13.8%)	2 (28.6%)	15 (11.6%)	16 (12.7%)	94 (14.3%)	
	2017	111 (12.1%)	2 (28.6%)	15 (11.6%)	16 (12.7%)	78 (11.9%)	
	2018	133 (14.5%)	0 (0.0%)	19 (14.7%)	18 (14.3%)	96 (14.6%)	
	2019	111 (12.1%)	2 (28.6%)	19 (14.7%)	12 (9.5%)	78 (11.9%)	
Sex, *n* (%)	Boys	683 (74.4%)	6 (85.7%)	75 (58.1%)	87 (69.0%)	515 (78.5%)	<0.001
Witness, *n* (%)	Arrests witnessed by bystanders	670 (73.0%)	5 (71.4%)	105 (81.4%)	105 (83.3%)	455 (69.4%)	0.001
First documented rhythm, *n* (%)	VF/pVT	17 (1.9%)	0 (0.0%)	2 (1.6%)	1 (0.8%)	14 (2.1%)	0.440
	PEA	316 (34.4%)	4 (57.1%)	53 (41.1%)	43 (34.1%)	216 (32.9%)	
	Asystole	585 (63.7%)	3 (42.9%)	74 (57.4%)	82 (65.1%)	426 (64.9%)	
Bystander CPR, *n* (%)		271 (29.5%)	4 (57.1%)	56 (43.4%)	52 (41.3%)	159 (24.2%)	<0.001
Advanced airway management, *n* (%)	ETI	18 (2.0%)	0 (0.0%)	0 (0.0%)	0 (0.0%)	18 (2.7%)	<0.001
	SGA	175 (19.1%)	0 (0.0%)	14 (10.9%)	10 (7.9%)	151 (23.0%)	
	Non	725 (79.0%)	7 (100.0%)	115 (89.1%)	116 (92.1%)	487 (74.2%)	
Intravascular fluid, *n* (%)		162 (17.6%)	0 (0.0%)	2 (1.6%)	13 (10.3%)	147 (22.4%)	<0.001
Epinephrine, *n* (%)		112 (12.2%)	0 (0.0%)	2 (1.6%)	8 (6.3%)	102 (15.5%)	<0.001
Call to contact with a patient by EMS, min, median (IQR)		9 (7–12)	9 (7–13)	9 (7–12)	8 (7–10)	9 (7–13)	0.007
Contact to hospital arrival, min, median (IQR)		24 (17–33)	25 (21–42)	22 (15–30)	21 (15–30)	25 (18–34)	<0.001
Daytime, *n* (%)		288 (31.4%)	4 (57.1%)	78 (60.5%)	68 (54.0%)	138 (21.0%)	<0.001
Weekend/holiday, *n* (%)		623 (67.9%)	3 (42.9%)	74 (57.4%)	96 (76.2%)	450 (68.6%)	0.005
One-month survival, *n* (%)		30 (3.3%)	0 (0.0%)	5 (3.9%)	8 (6.3%)	17 (2.6%)	0.160
Prehospital ROSC, *n* (%)		60 (6.5%)	0 (0.0%)	6 (4.7%)	13 (10.3%)	41 (6.2%)	0.230
Neurologically favorable outcome, *n* (%)		6 (0.7%)	0 (0.0%)	0 (0.0%)	2 (1.6%)	4 (0.6%)	0.460

EMS, emergency medical services; ETI, endotracheal intubation; IQR, interquartile range; SGA, supraglottic airway; PEA, pulseless electrical activity; VF, ventricular fibrillation; pVT, pulseless ventricular tachycardia; ROSC, return of spontaneous circulation.

**Table 2 jcm-11-00831-t002:** Annual trends in primary and secondary outcomes of traumatic out-of-hospital cardiac arrests after traffic collisions.

	2013	2014	2015	2016	2017	2018	2019	
	*n* = 162	*n* =1 40	*n* = 134	*n* = 127	*n* = 111	*n* = 133	*n* = 111	
**One-month survival, *n* (%)**	3 (1.9%)	5 (3.6%)	4 (3.0%)	3 (2.4%)	6 (5.4%)	4 (3.0%)	5 (4.5%)	OR for 1-year increment
Crude OR	reference	1.96	1.63	1.28	3.03	1.64	2.50	1.11
95%CI		(0.46–8.37)	(0.36–7.42)	(0.25–6.46)	(0.74–12.38)	(0.36–7.48)	(0.59–10.68)	(0.93–1.33)
Adjusted OR	reference	2.34	1.54	1.39	3.81	1.99	2.77	1.13
95%CI		(0.53–10.43)	(0.29–8.11)	(0.26–7.36)	(0.87–16.64)	(0.42–9.53)	(0.60–12.73)	(0.93–1.37)
**Prehospital ROSC, *n* (%)**	11 (6.8%)	7 (5.0%)	5 (3.7%)	12 (9.4%)	11 (9.9%)	5 (3.8%)	9 (8.1%)	OR for 1-year increment
Crude OR	reference	0.72	0.53	1.43	1.51	0.54	1.21	1.04
95%CI		(0.27–1.92)	(0.18–1.57)	(0.61–3.36)	(0.63–3.62)	(0.18–1.58)	(0.48–3.03)	(0.91–1.18)
Adjusted OR	reference	0.73	0.61	1.48	1.54	0.52	1.19	1.03
95%CI		(0.26–2.00)	(0.20–1.86)	(0.60–3.64)	(0.62–3.84)	(0.17–1.58)	(0.46–3.13)	(0.90–1.17)
**Neurologically favorable outcome, *n* (%)**	1 (0.6%)	0 (0.0%)	0 (0.0%)	1 (0.8%)	2 (1.8%)	1 (0.8%)	1 (0.9%)	OR for 1-year increment
Crude OR	reference	NA	NA	1.28	2.95	1.22	1.46	1.24
95%CI				(0.08–20.63)	(0.26–32.98)	(0.08–19.69)	(0.09–23.65)	(0.82–1.88)
Adjusted OR	reference	NA	NA	1.72	3.95	1.23	2.07	1.28
95%CI				(0.09–32.84)	(0.30–51.82)	(0.07–22.32)	(0.11–39.55)	(0.83–1.97)

OR = odds ratio; CI = confidence interval; ROSC = return of spontaneous circulation.

**Table 3 jcm-11-00831-t003:** Factors associated with one-month survival.

		All (*n*)	One-Month Survival (*n*)	(%)	Crude OR	95% CI	Adjusted OR	95% CI
Age group, *n* (%)	<1 year old	7	0	0.0%	NA		NA	
	1–5 years old	129	5	3.9%	(reference)		(reference)	
	6–11 years old	126	8	6.3%	1.68	(0.53–5.29)	1.99	(0.60–6.61)
	12–19 years old	656	17	2.6%	0.66	(0.24–1.82)	0.70	(0.22–2.29)
Sex, *n* (%)	Girls	235	7	3.0%	(reference)		(reference)	
	Boys	683	23	3.4%	1.14	(0.48–2.68)	1.62	(0.63–4.20)
Witness, *n* (%)	Arrests witnessed by bystanders	670	25	3.7%	1.88	(0.71–4.98)	1.52	(0.53–4.35)
	Arrests not witnessed	248	5	2.0%	(reference)		(reference)	
First documented rhythm, *n* (%)	VF/pVT	17	2	11.8%	8.53	(1.70–42.93)	12.15	(2.04–72.36)
	PEA	316	19	6.0%	4.09	(1.83–9.16)	4.46	(1.82–10.93)
	Asystole	585	9	1.5%	(reference)		(reference)	
Bystander CPR, *n* (%)	No	647	20	3.1%	(reference)		(reference)	
	Yes	271	10	3.7%	1.20	(0.55–2.60)	0.84	(0.36–1.98)
Advanced airway management, *n* (%)	ETI	18	0	0.0%	NA		NA	
	SGA	175	6	3.4%	1.04	(0.42–2.58)	1.32	(0.46–3.80)
	None	725	24	3.3%	(reference)		(reference)	
Intravascular fluid, *n* (%)	No	756	24	3.2%	(reference)		(reference)	
	Yes	162	6	3.7%	1.17	(0.47–2.92)	0.51	(0.08–3.28)
Epinephrine, *n* (%)	No	806	24	3.0%	(reference)		(reference)	
	Yes	112	6	5.4%	1.84	(0.74–4.62)	3.23	(0.51–20.37)
Call to contact with a patient by EMS (for one-minute increment)					0.96	(0.89–1.04)	1.01	(0.93–1.09)
Contact to hospital arrival (for one-minute increment)					0.97	(0.94–1.00)	0.98	(0.95–1.02)
Daytime, *n* (%)	No	630	16	2.5%	(reference)		(reference)	
	Yes	288	14	4.9%	1.96	(0.94–4.07)	1.36	(0.58–3.18)
Weekend/holiday, *n* (%)	No	872	28	3.2%	(reference)		(reference)	
	Yes	46	2	4.3%	1.37	(0.32–5.94)	1.92	(0.41–9.10)
Year of onset (for one-year increment)					1.11	(0.93–1.33)	1.13	(0.93–1.37)

ETI, endotracheal intubation; VF, ventricular fibrillation; pVT, pulseless ventricular tachycardia; SGA, supraglottic airway; PEA, pulseless electrical activity; OR, odds ratio; CI, confidence interval; CPR, cardiopulmonary resuscitation; EMS, emergency medical service; NA, not applicable.

**Table 4 jcm-11-00831-t004:** Factors associated with prehospital return of spontaneous circulation.

		All (*n*)	Prehospital ROSC (*n*)	(%)	Crude OR	95% CI	Adjusted OR	95% CI
Age group, *n* (%)	<1 year old	7	0	0.0%	NA		NA	
	1–5 years old	129	6	4.7%	(reference)		(reference)	
	6–11 years old	126	13	10.3%	2.36	(0.87–6.41)	2.53	(0.89–7.19)
	12–19 years old	656	41	6.3%	1.37	(0.57–3.29)	1.58	(0.60–4.19)
Sex, *n* (%)	Girls	235	16	6.8%	(reference)		(reference)	
	Boys	683	44	6.4%	0.94	(0.52–1.70)	0.91	(0.48–1.71)
Witness, *n* (%)	Arrests witnessed by bystanders	670	44	6.6%	1.02	(0.56–1.84)	0.91	(0.48–1.73)
	Arrests not witnessed	248	16	6.5%	(reference)		(reference)	
First documented rhythm, *n* (%)	VF/pVT	17	1	5.9%	1.86	(0.2–14.77)	1.29	(0.15–11.13)
	PEA	316	40	12.7%	4.32	(2.45–7.59)	3.96	(2.16–7.25)
	Asystole	585	19	3.2%	(reference)		(reference)	
Bystander CPR, *n* (%)	No	647	37	5.7%	(reference)		(reference)	
	Yes	271	23	8.5%	1.53	(0.89–2.63)	1.24	(0.68–2.24)
Advanced airway management, *n* (%)	ETI	18	1	5.6%	0.81	(0.11–6.23)	0.50	(0.06–4.38)
	SGA	175	10	5.7%	0.84	(0.41–1.69)	0.60	(0.27–1.36)
	None	725	49	6.8%	(reference)		(reference)	
Intravascular fluid, *n* (%)	No	756	43	5.7%	(reference)		(reference)	
	Yes	162	17	10.5%	1.94	(1.08–3.50)	1.14	(0.40–3.30)
Epinephrine, *n* (%)	No	806	45	5.6%	(reference)		(reference)	
	Yes	112	15	13.4%	2.62	(1.40–4.87)	2.24	(0.76–6.60)
Call to contact with a patient by EMS (for one-minute increment)					0.97	(0.92–1.02)	1.00	(0.95–1.05)
Daytime, *n* (%)	No	630	37	6.8%	(reference)		(reference)	
	Yes	288	23	2.2%	1.39	(0.81–2.39)	1.34	(0.72–2.47)
Weekend/holiday, *n* (%)	No	872	59	6.8%	(reference)		(reference)	
	Yes	46	1	2.2%	0.31	(0.04–2.26)	0.39	(0.05–3.00)
Year of onset (for one-year increment)					1.04	(0.91–1.18)	1.03	(0.90–1.17)

ROSC, return of spontaneous circulation; ETI, endotracheal intubation; VF, ventricular fibrillation; pVT, pulseless ventricular tachycardia; SGA, supraglottic airway; PEA, pulseless electrical activity; OR, odds ratio; CI, confidence interval; CPR, cardiopulmonary resuscitation; EMS, emergency medical service; NA, not applicable.

## Data Availability

The data that support the findings of this study are available from the All-Japan Utstein Registry; restrictions apply to the availability of these data, which were used under license for the current study, and they are therefore not publicly available. However, data are available from the authors upon reasonable request and with permission from the All-Japan Utstein Registry.

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
