# Peer review of "Epidemiology and Outcome of Pediatric Out-of-Hospital Cardiac Arrest after Traffic Collision in Japan: A Population-Based Study"

_jcm, 2022, doi:10.3390/jcm11030831_

Round 1
Reviewer 1 Report
I appreciate the authors for conducting a fascinating study in road traffic injury. The title and objective of the study are stated clearly. The method section is well described. The result and discussion were articulated according to the study objective.
The conclusion was made correctly. However, you can avoid the repetition of results in the conclusion section.
Author Response
Response to Reviewer 1:
I appreciate the authors for conducting a fascinating study in road traffic injury. The title and objective of the study are stated clearly. The method section is well described. The result and discussion were articulated according to the study objective.
The conclusion was made correctly. However, you can avoid the repetition of results in the conclusion section.
Response: Thank you for your insightful comment. As we also received a comment on the conclusion from reviewer 2, we have revised the section as follows (Lines 344-348) :
Original: The large OHCA registry in Japan demonstrated that 1-month survival after traumatic OHCAs following traffic collision was approximately 3%, and the survival trends did not improve year-by-year. However, some children even gain full recovery of neurological function. Prevention and further prehospital treatment of traumatic OHCAs following traffic collision are warranted to improve survival outcomes after OHCA in this group.
Revised: The large OHCA registry in Japan demonstrated that 1-month survival after traumatic OHCAs following traffic collision was poor, and the survival trends did not improve year-by-year. However, some children even gain full recovery of neurological function. Research related to better safety equipment for children and enhanced training for pediatric related trauma are warranted to improve survival outcomes after OHCA in this group.

Reviewer 2 Report
Dear authors,
thank you for the possibility to review this interesting scientific paper.
I have some remarks I would like to share with you:
ll. 68 ff: Can you explain why you chose the age limit of < 20 years. Pediatric patients after puberty in most cases can be treated similarly to adults, therefore the children aged starting about 15 and 19 could possibly weaken the results for the rest of the pediatric population. Did you try to look for results excluding this age group? I can only see the division of age groups you have provided (12-19 years).
ll. 78 ff.: diagnoses are made clinically. No post mortal CT scan or autopsy has supported this suspicion. Isn´t this a limitation of the study which should be mentioned later (ll. 323 ff.)?
l. 99: "dependent cyanosis". Maybe it is just me who is not aware of what this precisely stands for and why it is an exclusion criterion. Could you explain, please?
Figure 1: You have mentioned 22,518 cerbrovascular diseases. I do not understand why you could exclude such a big number. There are 7,475 cases of non-cardiac origin only, aren´t there?
ll. 281 ff.: How can you be sure that the accident was followed by CPA and not vice versa?
ll. 339-344: If this is your conclusion, should this not be followed by a call for research on why there has been no improvement over the last years, e.g. research for better equipment for children, better training for pediatric related trauma etc.?
Kind regards
